# Spin Entropy

**DOI:** 10.3390/e24091292

**Published:** 2022-09-14

**Authors:** Davi Geiger, Zvi M. Kedem

**Affiliations:** Courant Institute of Mathematical Sciences, New York University, New York, NY 10012, USA

**Keywords:** spin entropy, geometric quantization, entangled states

## Abstract

Two types of randomness are associated with a mixed quantum state: the uncertainty in the probability coefficients of the constituent pure states and the uncertainty in the value of each observable captured by the Born’s rule probabilities. Entropy is a quantification of randomness, and we propose a spin-entropy for the observables of spin pure states based on the phase space of a spin as described by the geometric quantization method, and we also expand it to mixed quantum states. This proposed entropy overcomes the limitations of previously-proposed entropies such as von Neumann entropy which only quantifies the randomness of specifying the quantum state. As an example of a limitation, previously-proposed entropies are higher for Bell entangled spin states than for disentangled spin states, even though the spin observables are less constrained for a disentangled pair of spins than for an entangled pair. The proposed spin-entropy accurately quantifies the randomness of a quantum state, it never reaches zero value, and it is lower for entangled states than for disentangled states.

## 1. Introduction

The quantification of randomness of a quantum state, via the entropy, gives the amount of *information* of a state (the inverse of entropy).

Von Neumann entropy has been an important concept applied to quantum computing. One important distinction between quantum computing and classical computing is the interference phenomena captured by entanglement of qubits. A much-studied challenge to building quantum computers is decoherence, where during computational processing, entangled states tend to disentangle by interacting with the environment, and the quantum interference is lost, e.g., see [1,2,3,4,5,6]. Thus, maintaining entanglement is a topic of much interest for constructing robust quantum algorithms and computers. To further understand these phenomena, an accurate quantification of the information flow during these processes may be necessary.

There are many similarities between a qubit and a spin of a particle. An arbitrary state for a qubit can be written as a linear combination of the Pauli matrices, which provide a basis for all 2×2 self-adjoint matrices. Thus, a straightforward connection between qubits and spin 12 formalism is established. However, for many physical models of qubits, for example using photon polarization states or an atom’s ground state vs. an excited state, the Pauli exclusion principle is removed. We exploit the similarities and dissimilarities between two spin 12 particles and two qubits when studying entanglement of two particles. Both cases require the understanding of spin randomness (information) and its entanglement information.

The entropy of the internal degrees of freedom (DOFs), that is, the spin of a particle, as well as the entropy of the spatial DOFs, applied to the evolution of quantum physical states may help our understanding of physics. We speculate that the evolution of physical systems requires the information content not to increase (the entropy not to decrease).

We examine previous efforts on quantifying randomness for a spin state and propose a new spin entropy. Extensions to qubits entropy for entanglement scenarios are also considered.

### 1.1. Previous Work

In classical physics, entropy was introduced to handle large ensembles of particles, where the complete specification of the state is not practical. In quantum physics, von Neumann entropy [7] quantifies only the lack of knowledge an observer has about a quantum state, i.e., it quantifies only the randomness in specifying its DOFs. Thus, for all pure single particles and for all pure entangled particles, von Neumann entropy is zero, even though only probabilities are knowable about the set of observables. The Stern–Gerlach experiment [8] illustrates the quantum scenario for spin, where spin 12 particles with a *z*-up direction state are prepared and, despite having zero von Neumann entropy, randomness in the spin values along any direction perpendicular to *z* is observed. Nevertheless, von Neumann entropy, with a limited evaluation of the randomness of a quantum system, has played an important role on our understanding of physical phenomena. The work on quantum thermalization [9,10,11,12] and their references suggests that the procedure of tracing out the environment and evaluating the reduced density matrix of a system of interest may relate to classical entropy. We argue that a complete quantification of randomness, including the randomness of the observable, will lead to a more accurate understanding of the role of entropy in physics.

Wehrl entropy [13] was introduced to approximate a classical entropy for a quantum state, and Lieb studied spin-coherent states to evaluate Wehrl spin entropy [14,15]. As in the case of spatial coherent states, spin coherent states constitute an overcomplete set of states. Due to this overcomplete representation, all spin 12 states, up to a global phase, are coherent states; in addition, a large set of spin 1 states are coherent states. Since Werhl entropy is based on an overcomplete basis representation, projections onto this basis lead to quasi-probabilities violating the Kolmogorov third axiom. The overcomplete basis decomposition and the arbitrariness of the choice of spin-coherent states basis to define the probability distribution prevent Wehrl entropy from accurately quantifying the randomness associated with the spin observables.

Those limitations of the von Neumann and the Wehrl entropies impact the evaluation of much-studied entangled states. For any pure entangled state, von Neumann entropy is zero, and when tracing out one particle of the entanglement, the von Neumann entropy of the resulting mixed state is larger than when tracing out the disentangled state, which leads to a single particle pure state with zero von Neumann entropy. Similarly, Wehrl entropy for any entangled state is also evaluated via tracing out the state, and it is larger for entanglement while being minimized by disentanglement. Note that Wehrl entropy is minimized for all spin-coherent states up to a phase [15,16].

### 1.2. Our Contribution

We propose a definition of spin-entropy to quantify the randomness of the spin observables once the DOFs of a quantum spin state have been specified. Our approach to precisely characterize randomness is to employ conjugate operators forming a spin phase space. Thus, we adopt the geometric quantization method [17,18,19]. In contrast with Wehrl entropy, the probabilities are derived from projections onto orthogonal basis functions, and thus they satisfy all the Kolmogorov axioms of probability. We show that the minimum of this entropy in phase space is ln2π due to the entropic uncertainty principle for conjugate spin operators. We extend the definition of spin-entropy to (i) mixed states, where classical randomness is assigned to each quantum state, (ii) quantum field theory (QFT), where the number of particles is also an observable, and (iii) qubits, where we consider the same spin-entropy for spin 12, in log base 2, without enforcing the Pauli exclusion principle for the quantum states.

We explore the much-studied topic of spin entanglements and disentanglements. For a two-spin system, we show that Bell entangled states, maximally entangled states, are assigned the lowest spin-entropy. We show that disentangled states are assigned maximum qubit-entropy. This is in contrast with previous quantum entropy work as pointed out earlier. We then propose that the degree of entanglement for any number of particles or any number of qubits be defined by the spin-entropy or qubit-entropy. The lower the entropy of an entangled state, the greater the entanglement. We also extend the analysis of the spin-entropy of entanglement states for the case of three fermions of spin 12.

We discuss possible directions to study thermalization via the proposed entropy. A list of symbols used is provided in Appendix B, in order of appearance.

## 2. Material and Methods: Spin Phase Space

We first briefly review geometric aspects of spin, including the geometric quantization method that leads to the spin phase space.

The spin matrix associated with a particle can be specified, e.g., [20], as $$\begin{array}{cc} \vec{S} & =S_{x}\hat{x}+S_{y}\hat{y}+S_{z}\hat{z}\;\mathrm{and}\;S^{2}=S_{x}^{2}+S_{y}^{2}+S_{z}^{2}\;, \end{array}$$
$$\begin{array}{cc} {}[S_{a},S_{b}] & =i\hbar S_{c}\;\;\mathrm{where}\;a,b,c\;\mathrm{is}\;a\;\mathrm{cyclic}\;\mathrm{permutation}\;\mathrm{of}\;x,y,z\;,\;\mathrm{and} \end{array}$$
$$\begin{array}{cc} {}[S^{2},S_{a}] & =0\;\;\mathrm{for}\;a=x,y,z\;. \end{array}$$The spin value of a particle is a Casimir invariant, but it is not possible to know the values of the spin projections in all directions in three dimensions, as knowing the value of the spin along the *z*-direction implies that the *x*- and the *y*- direction spin values are unknowable. This uncertainty reflects the close relation between spin matrices, their unitary transformations, and the rotation group SO(3). For spin 12 particles, any spin state is reachable from any other spin state via a 2×2 unitary transformation, which is a local isomorphism (and a global homomorphism) to the SO(3) group. For spin 1, the matrices are unitarily similar to SO(3), and they can be transformed into generators of SO(3) via unitary transformations.

Two observations about Geometric Quantization (GQ), which we describe next, lead us to adopting this method. One is the relevance of the SO(3) group to modeling spin states which leads to the quantizing of the sphere itself. The other is that, at any given time, a spin observable is the spin value along one chosen direction, say *z*-direction, and thus the uncertainty principle formed by a pair of conjugate variables is between the spin along a *z*-direction and not-the-*z*-direction. Note that a choice for the *x*- or *y*- axis direction is not required with a choice of *z*-direction.

### 2.1. The Geometric Quantization Approach to Spin Phase Space

The phase space of a spin is derived from quantizing the sphere as it is developed by the GQ method, e.g., see [17,18,19], and we summarize it now. The sphere is the surface of the ball with a radius of the spin magnitude sℏ.

On the sphere, the *z* values are specified by the polar representation sℏcosθ, while the values on the intersection with planes perpendicular to the *z*-axis are specified by the azimuth angle ϕ. Treating the sphere as a phase space, a rescaled symplectic 2-form in spherical polar coordinates is (dω=(dp∧(dq=sℏsinθ(dθ∧(dϕ, and so the Lagrangian is $L=p\dot{q}=s\hbar \cos\theta \;\dot{\phi }$, and the action is SL=∫L(dt=sℏ∫cosθ(dϕ. From this Lagrangian, the GQ method derives the uncertainty commutation relation
$$\begin{array}{cc} \left[\phi ,s\hbar \cos\theta \right] & =i\hbar \;, \end{array}$$
where the conjugate pair of eigenvalues (ϕ,sℏcosθ)∈[0,2π)×{−sℏ,⋯,sℏ} forms a spin phase space with a finite Hilbert space volume 2(2s+1)πℏ.

Note that the rotation operator $e^{-i\frac{S_{z}}{\hbar }\phi }$ of an angle ϕ around the *z*-axis describes the polarization angle in the *x*-*y* plane. Thus, we will refer to the angle ϕ as the polarization angle. In order to create physical quantities with the polarization operator ϕ, we must constrain ϕ to a periodic function with period 2π, i.e., to values that are a function of $e^{i\phi }$.

At the northern pole (cosθ=1) and the southern pole (cosθ=−1), the angle ϕ are not defined. Thus, in the basis |ϕ〉 that diagonalizes ϕ, we have two operators, namely $s\hbar \cos\theta =s\hbar -i\hbar \frac{}{\phi }$ for the northern hemisphere and $s\hbar \cos\theta =-s\hbar -i\hbar \frac{}{\phi }$ for the southern hemisphere.

In the basis |ϕ〉, the eigenstates of the operator $S_{z}=s\hbar \cos\theta$ are
|ξs,m〉=∫|ϕ〉〈ϕ||ξs,m〉(dϕ=∫ψs,m(ϕ)|ϕ〉(dϕ,
where m=−s,⋯,s and
(1)$$\psi_{s,m}\left(\phi \right)=\left\{\begin{array}{ccc} \frac{1}{\sqrt{2\pi }}e^{i(s+m)\phi }, & \;m\geq 0 & \;(\mathrm{northern}\;\mathrm{hemisphere})\;;\\ \frac{1}{\sqrt{2\pi }}e^{i(-s+m)\phi }, & \;m<0 & \;(\mathrm{southern}\;\mathrm{hemisphere})\;. \end{array}\right.$$The two solutions in (1) are periodic in ϕ and differ by a phase (gauge) transformation of $e^{-i2s\phi }$.

Consider a particle state $|\xi_{s}\rangle$ of spin magnitude sℏ. This state in the basis of the eigenvectors of $S_{z}=s\hbar \cos\theta$, and $S^{2}$ is
$$\begin{array}{c} |\xi_{s}\rangle =\sum_{m=-s}^{s}\alpha_{s,m}|\xi_{s,m}\rangle \;, \end{array}$$
where $\alpha_{s,m}=\langle \xi_{s,m}||\xi_{s}\rangle \in C$ sastisfying $1=\sum_{m=-s}^{s}{\left|\alpha_{s,m}\right|}^{2}$. In the basis of the conjugate variable ϕ, the state is
|ξs〉=∫02π|ϕ〉〈ϕ||ξs〉(dϕ=∫02π|ϕ〉∑m=−ssαs,m〈ϕ||ξs,m〉(dϕ
(2)=∫02π∑m=−ssαs,mψs,m(ϕ)|ϕ〉(dϕ=∫02πλs(ϕ)|ϕ〉(dϕ,
where
(3)$$\begin{array}{c} \lambda_{s}\left(\phi \right)=\langle \phi ||\xi_{s}\rangle =\sum_{m-s}^{s}\alpha_{s,m}\psi_{s,m}\left(\phi \right)\;. \end{array}$$Thus, for a state $|\xi_{s}\rangle$ with density matrix $\rho_{s}=|\xi_{s}\rangle \langle \xi_{s}|$, the probabilities of the phase space are the product of the probabilities $\left\{\rho_{s,m}=\langle \xi_{s,m}|\rho_{s}|\xi_{s,m}\rangle =|\alpha_{s,m}{|}^{2}\right\}$ with the probability densities $\left\{\rho_{s}\left(\phi \right)=\langle \phi |\rho_{s}|\phi \rangle =|\lambda_{s}\left(\phi \right){|}^{2}\right\}$.

### 2.2. The Spin-Entropy in Spin Phase Space

Our goal is to quantify the randomness given a spin state. In order to capture the randomness of the observables without double counting them, we specify the spin conjugate operators that create the phase space. The phase space captures all the randomness of the observables of a spin state.

We define the spin-entropy of a pure quantum state $|\xi_{s}\rangle$ with spin *s* in spin phase space to be

**Definition** **1.**

S=Sz+Sz⊥=−∑m=−ssρs,mlnρs,m−∫ρs(ϕ)lnρs(ϕ)(dϕ


(4)
=−∑m=−ss|αs,m|2ln|αs,m|2−∫|λs(ϕ)|2ln|λs(ϕ)|2(dϕ.



The first term is the Shannon entropy capturing the randomness of the spin value along the *z*-axis. The second term is differential entropy capturing the randomness of the spin value in the plane perpendicular to the *z*-axis, i.e., the entropy of the polarization angle ϕ. We define *intrinsic-spin-information*, represented by Γ, as the inverse of the entropy, i.e.,
Γ=−1∑m=−ss|αs,m|2ln|αs,m|2+∫|λs(ϕ)|2ln|λs(ϕ)|2(dϕ.
Γ is non-negative and quantifies what is known about the observables of a state, where infinity indicates full knowledge of all the observable values simultaneously.

We conjecture that the spin-entropy (4) depends only on the variables that define the spin-entropy’s component along the *z*-axis. This is further elaborated and utilized in Conjecture 1. We also conjecture that the states with maximum and minimum spin-entropy values are the ones with maximum and minimum spin-entropy component along the *z*-axis.

### 2.3. Illustrative Example

To illustrate the process of evaluating the spin-entropy, consider the case of a state with $\alpha_{s,m}=\sqrt{2\pi \;/\;2s+1}\;\psi_{s,m}^{*}\left(\phi_{0}\right)$, where $\phi_{0}$ is an arbitrary given phase value. This state has a uniform distribution $|\alpha_{s,m}{|}^{2}=1\;/\;2s+1$, i.e., it is a state with the lowest intrinsic-spin-information along the *z*-axis. Then,
$$\begin{array}{cc} \lambda_{s}\left(\phi \right) & =\frac{1}{\sqrt{2\pi \;2s+1}}\sum_{m=-s}^{s}e^{i\left((2\theta (m)-1)\;s+m\right)\left(\phi -\phi_{0}\right)} \end{array}$$
$$\begin{array}{cc} \; & =\frac{1}{\sqrt{2\pi \;(2s+1)}}\left(1-\mathrm{mod}\left(2s,2\right)+2\sum_{j=j_{\min}}^{2s}\cos\left(j\;(\phi -\phi_{0})\right)\right)\;, \end{array}$$
where $j_{\min}=s+1-\frac{1}{2}\mathrm{mod}\left(2s,2\right)$, and mod is the modulus function. Note that the distribution $\lambda_{s}\left(\phi \right)$ becomes more concentrated around $\phi_{0}$ as *s* increases. The intrinsic-spin-information about the spin on the plane $z^{⊥}$ increases as *s* increases. However, as *s* increases, the total intrinsic-spin-information decreases because of the increase in spin-entropy along *z*. For a fermion of spin 12, the spin-entropy becomes
S=ln2+lnπ−1π∫02πcos2(ϕ−ϕ0)lncos2(ϕ−ϕ0)(dϕ
$$\begin{array}{cc} \; & =\ln2+\ln\pi +\ln\frac{4}{e}\approx 1.531+\ln2\;, \end{array}$$
with Γ ≈0.450. For spin 1, S≈1.270+ln3 with Γ ≈0.422.

### 2.4. Minimum Spin-Entropy

The third law of thermodynamics establishes zero as the minimum of thermodynamics entropy. The use of differential entropy for the term $S_{s,\phi }^{z^{⊥}}$ may suggest that the proposed entropy need not be positive, but we will establish a positive minimum for it.

In quantum mechanics, ignoring the spin and focusing on the spatial DOFs, the entropic uncertainty principle [21,22,23] establishes that $S_{x}+S_{p}\geq 3\ln\left(e\hbar \pi \right)$, with equality for the normal distribution. We now derive a bound for the spin-entropy.

**Theorem** **1.**
*The spin-entropy satisfies the inequality*

(5)
$$\begin{array}{c} S\geq \ln2\pi \;, \end{array}$$

*with equality attained for the eigenstates of the spin operator $S_{z}=s\hbar \cos\theta$.*


**Proof.** This proof is an adaptation to the specific spin phase space of the work of [21,22,23].In Appendix A, we prove Lemma A1 that shows that the Hausdorff–Young inequality holds for the spin states phase space. Thus, we can apply Beckner’s theorem [22]
1qn2q1p−n2p≥∑m=−ss|αs,m|q1q(2π)p−22∫02π|λs(ϕ)|p(dϕ1p,
where 1<p≤2, $\frac{1}{p}+\frac{1}{q}=1$, and *n* is the dimensionality of the space in which the functions are defined. For the sphere, n=2. Applying the logarithm to both sides of the inequality and noting that all the quantities are non-negative, we obtain
(6)ln1q1q1p−1p+ln(2π)p−22∫02π|λs(ϕ)|p(dϕ1p≥ln∑m=−ss|αs,m|q1q.Substituting p=2α and q=2β, we get the constraint α/1−α=−β/1−β∈(1,∞) from 1=1p+1q. Multiplying the terms on both sides of (6) by either one of these equal and positive expressions α1−α or −β1−β, and rearranging terms, we obtain
11−αln∫02π|λs(ϕ)|2α(dϕ+11−βln∑m=−ss|αs,m|2β≥11−αln12α+11−βln12β+ln2π,
where the two first terms are the Rényi’s differential entropy and the Rényi’s entropy, forming an entropic inequality. Thus, by taking the limits $\alpha \to 1^{-}$ and $\beta \to 1^{+}$, we obtain (5).The minimum entropy is reached for the eigenstates $|\xi_{s,m_{0}}\rangle$ of $S_{z}$, where $-s\leq m_{0}\leq s$. Then, $\alpha_{s,m}=\langle \xi_{s,m}||\xi_{s,m_{0}}\rangle =\delta_{m,m_{0}}$, and so $\lambda_{s}\left(\phi \right)=\psi_{s,m_{0}}$. Then, the probabilities are $|\alpha_{s,m}{|}^{2}=\delta_{m,m_{0}}$ and probability densities are $|\lambda_{s}{\left(\phi \right)|}^{2}=1\;/\;2\pi$, and so the Shannon entropy associated with $\left\{\alpha_{s,m}\right\}$ is zero and the differential entropy associated with $\left\{\lambda_{s}\left(\phi \right)\right\}$ is ln2π. □

Thus, for the intrinsic-spin-information, 0≤Γ≤1ln2π.

Two observations:The spin-entropic principle inequality for a bounded variable ϕ and an integer variable *m* resulting from the Fourier series transformation reaches its minimum for the eigenstates of the $S_{z},S^{2}$ operators, with corresponding $S_{z}$ eigenvalues *m*’s.Preparing a system to align a spin state with a particular direction, an eigenstate $|\xi_{s,m=s}\rangle$ of a *z*-axis, as it is done in many experiments with spin up along *z*, provides the knowledge of the spin along that direction. Thus, the intrinsic-spin-information is gained by such a preparation of the lowest spin-entropy state.

### 2.5. Extending the Spin-Entropy to Mixed States

Mixed states extend the Hilbert space of specified quantum states, or pure states, to quantum states that are not fully specified and instead are described by a classical probabilistic combination of pure states. Let a spin mixed state be defined via the density matrix
$$\begin{array}{c} \rho_{s}^{M}=\sum_{i=1}^{P}\gamma_{i}|\xi_{s}^{i}\rangle \langle \xi_{s}^{i}|\;, \end{array}$$
where $|\xi_{s}^{i}\rangle =\sum_{m=-s}^{s}\alpha_{s,m}^{i}|\xi_{s,m}\rangle ,\;i=1,\cdots ,P$ are pure states. The randomness present in a pure state is concentrated in the observables of the spin phase space. However, for mixed states, additional randomness exists in specifying the state itself, and it is captured by the probabilities $\gamma_{i}$. Then, the coefficients of a projection of each pure state density matrix onto the eigenstates along the *z*-axis choice or along the $z^{⊥}-|\phi \rangle$ direction are
$$\begin{array}{cc} \rho_{s,m,i}^{\gamma } & =\gamma_{i}\;\langle \xi_{s,m}||\xi_{s}^{i}\rangle \langle \xi_{s}^{i}||\xi_{s,m}\rangle =\gamma_{i}\;{\left|\alpha_{s,m}^{i}\right|}^{2}\;, \end{array}$$
$$\begin{array}{cc} \rho_{s,i}^{\gamma }\left(\phi \right) & =\gamma_{i}\;\langle \phi ||\xi_{s}^{i}\rangle \langle \xi_{s}^{i}||\phi \rangle =\gamma_{i}{\left(\sum_{m^{\prime }=-s}^{s}\alpha_{s,m^{\prime }}^{i}\psi_{s,m^{\prime }}\left(\phi \right)\right)}^{*}\sum_{m=-s}^{s}\alpha_{s,m}^{i}\psi_{s,m}\left(\phi \right) \end{array}$$
with the normalizations $1=\sum_{i=1}^{N}\sum_{m=-s}^{s}\rho_{s,m,i}^{\gamma }$ and 1=∑i=1N∫ρs,iγ(ϕ)(dϕ. Thus, extending (4) to include mixed states, the spin-entropy is then
(7)SM=−∑i=1N∑m=−ssρs,m,iγlnρs,m,iγ−∑i=1N∫ρs,iγ(ϕ)lnρs,iγ(ϕ)(dϕ=SvN+∑i=1NγiSsi,
where von Neumann entropy is $S^{\mathrm{vN}}=-\sum_{i=1}^{N}\gamma_{i}\ln\gamma_{i}$, and Ssi=−∑m=−ss|αs,mi|2ln|αs,mi|2+∫λsi(ϕ)lnλsi(ϕ)(dϕ is the entropy of each pure state in the mixture. The spin-entropy is always larger than von Neumann entropy, since we also consider the randomness of the observables, and for every *i*, $S_{s}^{i}\geq 0$, and so $\sum_{i=1}^{N}\gamma_{i}S_{s}^{i}\geq 0$.

Note that, if one considers only the randomness of the observables, the distributions in spin phase space are
$$\begin{array}{cc} \rho_{s,m}^{M} & =\langle \xi_{s,m}|\rho^{M}|\xi_{s,m}\rangle =\sum_{i=1}^{N}\gamma_{i}\;{\left|\alpha_{s,m}^{i}\right|}^{2}\;, \end{array}$$
$$\begin{array}{cc} \rho_{s}^{M}\left(\phi \right) & =\langle \phi |\rho^{M}|\phi \rangle =\sum_{i=1}^{N}\gamma_{i}{\left(\sum_{m^{\prime }=-s}^{s}\alpha_{s,m^{\prime }}^{i}\psi_{s,m^{\prime }}\left(\phi \right)\right)}^{*}\sum_{m=-s}^{s}\alpha_{s,m}^{i}\psi_{s,m}\left(\phi \right)\;. \end{array}$$
with normalizations $1=\sum_{m=-s}^{s}\rho_{s,m}^{M}$ and 1=∫ρsM(ϕ)(dϕ.

### 2.6. Pure System with Multiple Particles and Quantum Field Theory

Consider the spin of a pure system with *N* particles of the same species, each with spin *s*. When all the *N* particles are aligned, the system spin is maximized at $s_{\max}=Ns$. For a state with spin value $s_{\max}$, the possible *z* values are $m=-s_{\max},\cdots ,s_{\max}$. However, for the system with *N* particles, all different values $s^{\prime }\in \left[s_{\min},s_{\max}\right]$ can occur, where $s_{\min}=s\;\mathrm{mod}\left(N,2\right)$ for fermions and $s_{\min}=0$ for bosons. For each $s^{\prime }$, all possible *z* components must be considered, so $m_{s^{\prime }}=-s^{\prime },\cdots ,s^{\prime }$. The eigenstate basis of the $S_{z},S^{2}$ operators associated with the system is then
$$\begin{array}{cc} \left\{|\xi^{N,s}\rangle \right\} & =\cup_{s^{\prime }=s_{\min}}^{s_{\max}}\cup_{m=-s^{\prime }}^{s^{\prime }}|\xi_{s^{\prime },m}\rangle \;. \end{array}$$For example, two fermions with s=12 will produce four *z* states, namely three states with $s^{\prime }=1$ and $m^{\prime }=-1,0,1$, and the fourth state with $s^{\prime }=0$ and $m^{\prime }=0$. Thus, the eigenstate basis of $S_{z},S^{2}$ operators for two fermions is $\left\{|\xi_{1,1}\rangle ,|\xi_{1,0}\rangle ,|\xi_{1,-1}\rangle ,|\xi_{0,0}\rangle \right\}$. In order to extend the spin-entropy to *N* particles of the same species, we first describe the density matrix of a state as
(8)$$\begin{array}{cc} \rho^{N,s} & =|\xi^{N,s}\rangle \langle \xi^{N,s}|=\left(\sum_{s^{\prime }=s_{\min}}^{s_{\max}}\sum_{m_{s}=-s^{\prime }}^{s^{\prime }}\alpha_{s^{\prime },m_{s}}|\xi_{s^{\prime },m_{s}}\rangle \right){\left(\sum_{s^{\prime \prime }=s_{\min}}^{s_{\max}}\sum_{m_{s}^{\prime }=-s^{\prime \prime }}^{s}\alpha_{s^{\prime \prime },m_{s}^{\prime }}\langle \xi_{s^{\prime \prime },m_{s}^{\prime }}|\right)}^{*}\\ \; & =\sum_{s^{\prime }=s_{\min}}^{s_{\max}}\sum_{m_{s}=-s^{\prime }}^{s^{\prime }}\sum_{s^{\prime \prime }=s_{\min}}^{s_{\max}}\sum_{m_{s}^{\prime }=-s^{\prime \prime }}^{s^{\prime \prime }}\alpha_{s^{\prime },m_{s}}\alpha_{s^{\prime \prime },m_{s}^{\prime }}^{*}|\xi_{s^{\prime },m_{s}}\rangle \langle \xi_{s^{\prime \prime },m_{s}^{\prime }}|\;. \end{array}$$Thus, projecting (8) onto the *z*-basis and onto the |ϕ〉 basis, we derive the density functions in spin phase space for $s^{\prime }\in \left[s_{\min},s_{\max}\right]$ as
$$\begin{array}{cc} \rho_{s^{\prime },m}^{N,s}=\langle \xi_{s^{\prime },m}|\rho^{N,s}|\xi_{s^{\prime },m}\rangle & =|\alpha_{s^{\prime },m}{|}^{2}\;\;\mathrm{for}\;m\in \left[-s^{\prime },s^{\prime }\right],\;\mathrm{and} \end{array}$$
$$\begin{array}{cc} \rho_{s^{\prime }}^{N,s}\left(\phi \right)=\langle \phi |\rho^{N,s}|\phi \rangle & =|\lambda_{s^{\prime }}{\left(\phi \right)|}^{2}=\sum_{m=-s^{\prime }}^{s^{\prime }}\sum_{m^{\prime }=-s^{\prime }}^{s^{\prime }}\alpha_{s^{\prime },m}\alpha_{s^{\prime },m^{\prime }}^{*}\;\psi_{s^{\prime },m}\left(\phi \right)\psi_{s^{\prime },m^{\prime }}^{*}\left(\phi \right)\;. \end{array}$$Then, the spin-entropy (4) of the system is
(9)SN,s=SN,sz+SN,sz⊥=−∑s′=sminsmax∑m=−s′s′ρmN,slnρs′,mN,s−∑s′=sminsmax∫ρs′N,s(ϕ)lnρs′N,s(ϕ)(dϕ=−∑s′=sminsmax∑m=−s′s′|αs′,m|2ln|αs′,m|2−∫|λs′(ϕ)|2ln|λs′(ϕ)|2(dϕ.
where the normalization is $1=\sum_{s^{\prime }=s_{\min}}^{s_{\max}}\sum_{m=-s^{\prime }}^{s^{\prime }}\rho_{s^{\prime },m}^{N,s}$ and 1=∑s′=sminsmax∫|λs′(ϕ)|2(dϕ, and the phase space describing a system of *N* particles of spin *s* consists of all the spheres with radiuses ℏs, for $s\in [s_{\min},s_{\min}+1,\cdots ,s_{\max}-1,s_{\max}]$.

In quantum field theory (QFT), superpositions of states with any number of particles of spin species are also considered. One can write a state as
$$\begin{array}{c} |\xi_{\mathrm{QFT}}\rangle =\sum_{N=1}^{\infty }\gamma_{N}\sum_{s^{\prime }=s_{\min}}^{s_{\max}}\sum_{m=-s^{\prime }}^{s^{\prime }}\alpha_{s^{\prime },m}^{N}|\xi_{s^{\prime },m}\rangle \;, \end{array}$$
where $\sum_{N=1}^{\infty }{\left|\gamma_{N}\right|}^{2}=1$, and both $s_{\min}$ and $s_{\max}$ depend on *N* and *s*. Due to the orthogonality of the states with different number of particles, it is straightforward to extend (9) to obtain
$$\begin{array}{cc} S_{\mathrm{QFT},s} & =\sum_{N=1}^{\infty }|\gamma_{N}{|}^{2}\left(S_{N,s}^{z}+S_{N,s}^{z^{⊥}}\right)-\sum_{N=1}^{\infty }|\gamma_{N}{|}^{2}\ln{\left|\gamma_{N}\right|}^{2}\;. \end{array}$$Thus, in QFT, the role of the magnitude square of the complex valued coefficient $\gamma_{N}$ resembles the mixed states coefficients to the entropy (7).

### 2.7. The z-Axis

The formulation of a spin operator and corresponding eigenstates requires a choice of a *z*-axis. This is evident from the spin phase space where the quantization of the spin is along the *z*-axis, and perpendicular to it a continuous polarization variable ϕ∈[0,2π) is assigned. A choice of *z*-axis must be made to construct the quantum state and to assign to it a unique entropy. Approaches such as [15] average their proposed entropy over all possible 3D rotations to eliminate the *z*-axis bias, ending up with a spin-entropy that bundles a large set of quantum states into a single entropy value. Instead, we investigate a physical property that causes the break of the isotropy of the 3D space for spin states. For the Stern–Gerlach (SG) experiment, the direction of the applied magnetic field does define the *z*-axis, splitting spins according to a positive $z^{+}$ or negative $z^{-}$ direction eigenstates.

For a system of particles with total spin *S*, its spin Hamiltonian can be described by $H=\frac{q\hbar }{2m}\left(gB_{e}+\left(g-1\right)B_{I}\right)\;\cdot \;S$, where *S* is the spin vector along an arbitrary direction, “·” is the inner product, *g* is the gyromagnetic ratio, *q* the charge of the system, *m* its inertial mass, $B_{e}$ the external magnetic field applied to the spin system, and $B_{I}=\frac{1}{c}E\times v\;/\;\sqrt{1-\frac{v^{2}}{c^{2}}}$ is the magnetic field in the rest frame of the system when the charged system is moving in an electric field *E* with velocity *v*. Thus, the preferred direction that breaks the isotropy of the 3D space and defines the *z*-axis as $gB_{e}+\left(g-1\right)B_{I}$.

Consider the example of photon emission by an excited hydrogen atom in state $2p_{z}\;\left(n=2,l=1,m=1\right)$ transitioning to state 1s(n=1,l=0,m=0). The main spin contribution to the Hamiltonian is through the spin orbit interaction, given by a term proportional to L·S, where *L* is the angular momentum of the electron. Thus, the *z*-axis is defined by the angular momentum direction, with possible quantum states associated with the numbers m=1,0,−1, i.e., the excited state angular momentum already broke the isotropy, and consequently the angular momentum conservation leads to the emission of a photon on a plane perpendicular to this *z*-axis.

### 2.8. Thermalization

A natural question arises, “Is there a link between such entropy and thermodynamics ?”. In recent years, ideas of thermalization coming from quantum mechanics arose in the works of [9,10,11,12] and their references. They are all similarly rooted in von Neumann density matrix for mixed states, though using different techniques. They considered a pure state of the universe to be formed by an entanglement of a system state and an environment state. Tracing out the environment states leads to a mixture of states for the system state. A consideration of typicality states leads to a link between the von Neumann entropy of this system state and thermodynamics entropy. Clearly, the same procedure can be applied to our proposed approach but such mixed state for the system would also contribute to the entropy with the randomness of the observables as described by (7). Perhaps, another process that does not require tracing out the environment state and maintains the whole system pure state can also be devised. In this article, we do not develop this topic further as we focus here on the spin-entropy for the cases of one, two, and three particles including the study of entanglement. Extending this entropy to a large number of particles and adopting possibly the thermalization approach to this entropy and to link it to thermodynamics entropy is left for future research.

## 3. Results

### 3.1. Spin-Entropy for One Particle

We first analyze the spin-entropy for a fermion with spin value 12, then we analyze a massive boson with spin 1, and we conclude this section by analyzing the photon entropy.

### 3.2. Spin 12

A spin state of a particle with spin 12, represented by a set of two orthonormal eigenstates |+〉=|ξ12,12〉=(1,0)T and |−〉=|ξ12,−12〉=(0,1)T of the operators $\left(S^{2},S_{z}\right)$ with associated eigenvalues s=12,m=±12, is
(10)|ξ12,z〉=eiφeiνcosθα|+〉+sinθα|−〉=eiφeiνcosθα,sinθαT
with $\theta_{\alpha }\in \left[0,\frac{\pi }{2}\right]$ and φ,ν∈[0,π).

**Proposition** **1**(spin-entropy for s=12). *The spin-entropy of a spin state with s=12, described by (10), is*
(11)S12(θα)=−cos2θαlncos2θα−sin2θαlnsin2θα+ln2π−12π∫02π1+sin2θαcos2ϕln1+sin2θαcos2ϕ(dϕ,

**Proof.** A state |ξ12〉 assigns the probability distribution along the *z*-axis eigenstates $P_{z}={\left(\begin{array}{cc} {\left|\langle +||\xi_{1\;/\;2}\rangle \right|}^{2}, & {\left|\langle -||\xi_{1\;/\;2}\rangle \right|}^{2} \end{array}\right)}^{T}={\left(\begin{array}{cc} \cos^{2}\theta_{\alpha }, & \sin^{2}\theta_{\alpha } \end{array}\right)}^{T}$. Thus, from (2) and (3), we obtain
|ξ1/2,ϕ〉=∫eiφ12πeiνeiϕcosθα+e−iϕsinθα|ϕ〉(dϕ,
and so $\rho_{\theta_{\alpha },\nu }\left(\phi \right)=\frac{1}{2\pi }\left(1+\sin2\theta_{\alpha }\cos\left(2\phi +\nu \right)\right)$. □

For a visualization of the entropy, see Figure 1.

Note that the Wehrl’s entropy for any state |ξ12〉 is a constant since all spin 12 states are identical up to a phase [14,15], yielding the same entropy.

### 3.3. Spin 1

For massive particles with spin s=1, the spin matrices are
$$S_{x}=\frac{1}{\sqrt{2}}\left(\begin{array}{ccc} 0 & 1 & 0\\ 1 & 0 & 1\\ 0 & 1 & 0 \end{array}\right)\;\;S_{y}=\frac{1}{\sqrt{2}}\left(\begin{array}{ccc} 0 & -i & 0\\ i & 0 & -i\\ 0 & i & 0 \end{array}\right)\;\;S_{z}=\left(\begin{array}{ccc} 1 & 0 & 0\\ 0 & 0 & 0\\ 0 & 0 & -1 \end{array}\right)\;\;S^{2}=\left(\begin{array}{ccc} 2 & 0 & 0\\ 0 & 2 & 0\\ 0 & 0 & 2 \end{array}\right)\;$$
yielding a basis representation formed with the eigenvectors of $S_{z}$ and $S^{2}$: ${|↑\rangle }_{z}=|\xi_{1,1}\rangle ={\left(\begin{array}{ccc} 1, & 0, & 0 \end{array}\right)}^{T}$, ${|\to \rangle }_{z}=|\xi_{1,0}\rangle ={\left(\begin{array}{ccc} 0, & 1, & 0 \end{array}\right)}^{T}$, ${|↓\rangle }_{z}=|\xi_{1,-1}\rangle ={\left(\begin{array}{ccc} 0, & 0, & 1 \end{array}\right)}^{T}$. A general state of spin s=1 in the basis aligned with the *z*-axis is
(12)$$\begin{array}{cc} |\xi_{z}\rangle & =e^{i\varphi_{y}}\left(\sin\theta_{\alpha }\cos\theta_{\beta }\;e^{i\varphi_{x}}{|↑\rangle }_{z}+\cos\theta_{\alpha }{|\to \rangle }_{z}+\sin\theta_{\alpha }\sin\theta_{\beta }\;e^{i\varphi_{z}}{|↓\rangle }_{z}\right)\;, \end{array}$$
with $\theta_{\alpha },\theta_{\beta }\in \left[0,\frac{\pi }{2}\right]$ and $\varphi_{x},\varphi_{z},\varphi_{y}\in \left[0,2\pi \right)$. Following (2), we write the state in the |ϕ〉 basis as
(13)|ξz⊥〉=eiφy2π∫sinθαcosθβei(φx+2ϕ)+cosθαeiϕ+sinθαsinθβei(φz−2ϕ)|ϕ〉(dϕ.

**Proposition** **2.**
*The spin-entropy of a spin state with s=1 in a state given by (12) is*

(14)
S1=−∫02πρ1(ϕ,φx,φz,θα,θβ)lnρ1(ϕ,φx,φz,θα,θβ)(dϕ+Sc(cos2θα)+sin2θαSc(cos2θβ),

*where*

$$\begin{array}{cc} S_{c}\left(\cos^{2}\theta \right) & =-\cos^{2}\theta \ln\left(\cos^{2}\theta \right)-\left(1-\cos^{2}\theta \right)\ln\left(1-\cos^{2}\theta \right)\\ \rho_{1}\left(\phi \right) & =\frac{1}{2\pi }|\sin\theta_{\alpha }\cos\theta_{\beta }\;e^{i(\varphi_{x}+2\phi )}+\cos\theta_{\alpha }e^{i\phi }+\sin\theta_{\alpha }\sin\theta_{\beta }\;e^{i(\varphi_{z}-2\phi )}{|}^{2}\\ \; & =\frac{1}{2\pi }[1+\sin^{2}\theta_{\alpha }\sin\left(2\theta_{\beta }\right)\cos\left(4\phi +\varphi_{x}-\varphi_{z}\right)\\ \; & \;+\sin\left(2\theta_{\alpha }\right)(\cos\theta_{\beta }\cos\left(\phi +\varphi_{x}\right)+\sin\theta_{\beta }\cos\left(3\phi -\varphi_{z}\right)]\;. \end{array}$$



**Proof.** Computing the three probabilities associated with state $|\xi_{z}\rangle$ and deriving its entropy term $S_{1}^{z}$ yields
$$\begin{array}{cc} S_{1}^{z}\; & =-\cos^{2}\theta_{\alpha }\ln\cos^{2}\theta_{\alpha }-\sin^{2}\theta_{\alpha }\left[\ln\sin^{2}\theta_{\alpha }-S_{c}\left(\cos^{2}\theta_{\beta }\right)\right]\;. \end{array}$$The entropy term $S_{1}^{z^{⊥}}$ is derived from the probability density associated with the state $|\xi_{z^{⊥}}\rangle$ from (13), and so
$$\begin{array}{cc} \rho_{1}\left(\phi \right) & =\frac{1}{2\pi }|\sin\theta_{\alpha }\cos\theta_{\beta }\;e^{i(\varphi_{x}+2\phi )}+\cos\theta_{\alpha }e^{i\phi }+\sin\theta_{\alpha }\sin\theta_{\beta }\;e^{i(\varphi_{z}-2\phi )}{|}^{2}\;. \end{array}$$□

As observed in simulations and in the special case (11), we conjecture that

**Conjecture** **1.**
*The spin-entropy depends only on the variables that define the spin-entropy component along the z-axis. In particular, the spin-entropy (14) of a state given by (12) can be simplified to*

(15)
S1(θα,θβ)=−∫02πρ(ϕ,θα,θβ)lnρ(ϕ,θα,θβ)(dϕ+Sc(cos2θα)+sin2θαSc(cos2θβ),

*where*

$$\begin{array}{cc} \rho (\phi ,\theta_{\alpha },\theta_{\beta }) & =\frac{1}{2\pi }|\sin\theta_{\alpha }\cos\theta_{\beta }\;e^{i2\phi }+\cos\theta_{\alpha }e^{i\phi }+\sin\theta_{\alpha }\sin\theta_{\beta }\;e^{-i2\phi }{|}^{2}\;. \end{array}$$



By Theorem 1, the spin-entropy is minimized at $\theta_{\alpha }=0$, at $\left(\theta_{\alpha }=\pi \;/\;2,\theta_{\beta }=0\right)$, and at $\left(\theta_{\alpha }=\pi \;/\;2,\theta_{\beta }=\pi \;/\;2\right)$, the three eigenstates along the *z*-axis. Thus, preparing a spin state orientation by aligning it with an axis reduces the entropy. The entropy given by (15) is visualized in Figure 2.

In order to compare (15) to Wehrl’s spin-entropy, we consider the spin 1 coherent states
(16)$$\begin{array}{cc} |s,\alpha \rangle & ={\left(\cos\frac{\theta }{2}\right)}^{2}|1,-1\rangle +\sqrt{2}\cos\frac{\theta }{2}\sin\frac{\theta }{2}e^{i\phi }|1,0\rangle +{\left(\sin\frac{\theta }{2}\right)}^{2}e^{i2\phi }|1,1\rangle \;. \end{array}$$The cases where θ is 0 or π show that the states |1,−1〉 and |1,1〉 are coherent states, and thus minimize the Wehrl spin-entropy. However, state |1,0〉 cannot be a coherent state and therefore it has higher Wehrl’s spin-entropy. All eigenstates of $S_{z}$ minimize the spin-entropy. Moreover, we note that coherent states (16) correspond to the general spin 1 state (12) via the mapping
$$\begin{array}{c} \phi =\varphi_{z}=-\varphi_{x}\;,\;\sin\theta =\sqrt{\frac{2\sin2\theta_{\beta }}{1+\sin2\theta_{\beta }}}=\sqrt{2}\cos\theta_{\alpha }\;. \end{array}$$There is a one-to-one mapping between θ and $2\theta_{\beta }$, both in the range [0,π]. Thus, changes in the angle $\theta_{\alpha }$ from the constraint $\cos\theta_{\alpha }=\sqrt{\sin2\theta_{\beta }\;/\;1+\sin2\theta_{\beta }}$ will increase Wehrl’s entropy above its minimum.

### 3.4. Photon Entropy

Photon is a massless spin 1 particle. Its group representation is ISO(2) or E(2), with the gauge transformation accounting for the transverse direction. This results in a particle with helicity ±1 and a two-dimensional polarization field propagating in the helicity direction or against it. Formally, one creates two circular polarized states, a right-hand side, $|h_{+}\rangle$, and a left-hand side, $|h_{-}\rangle$, which are the quantum states of the polarization field. Given an x,y,z coordinate system, we write $|h_{+}\rangle =\frac{1}{\sqrt{2}}{\left(1,i\right)}^{T}$ and $|h_{-}\rangle =\frac{1}{\sqrt{2}}{\left(1,-i\right)}^{T}$ in the basis $|x\rangle ={\left(1,0\right)}^{T},\;|y\rangle ={\left(0,1\right)}^{T}$ in the plane perpendicular to *z*. The photon-spin operator along *z* is then
$$\begin{array}{c} S_{z}=\hbar \left(|h_{+}\rangle \langle h_{+}|-|h_{-}\rangle \langle h_{-}|\right)=\hbar \left(\begin{array}{cc} 0 & -i\\ i & 0 \end{array}\right)\;, \end{array}$$
with the two eigenstates, $|h_{+}\rangle ,|h_{-}\rangle$ and eigenvalues ±1, which are the helicity values. The general state of the photon is then $|\psi \rangle =\alpha_{+}|h_{+}\rangle +\alpha_{-}|h_{-}\rangle$ where $1=|\alpha_{+}{|}^{2}+{\left|\alpha_{-}\right|}^{2}$. The DOFs of the photon spin are captured by these two coefficients, which produce the probabilities $|\alpha_{+}{|}^{2},\;{\left|\alpha_{-}\right|}^{2}$ for the photon to be in each of the helicities eigenvectors.

### 3.5. Entanglement and Qubits Entropy

An entangled state is a quantum state that cannot be factored as a product of states of its local constituents. We study entanglements of two fermions, perform comparisons with von Neumann entropy and Wehrl entropy, and study entanglement of three fermions (triplets).

#### Two Fermions

Consider a system with two fermions with spin s=12, say *A* and *B*, and a spin basis, which is the product of individual spin eigenstates along the *z*-axis:$$\begin{array}{cc} |++\rangle ={|+\rangle }^{A}{|+\rangle }^{B}, & \;\;|+-\rangle ={|+\rangle }^{A}{|-\rangle }^{B}, \end{array}$$$$\begin{array}{cc} |-+\rangle ={|-\rangle }^{A}{|+\rangle }^{B}, & \;\;|--\rangle ={|-\rangle }^{A}{|-\rangle }^{B}. \end{array}$$In order to evaluate the spin-entropy of a two spin state, we examine the spin phase space for the two fermions case. Spin matrices associated with the two fermions can be written in terms of Pauli matrices $\sigma_{x},\sigma_{y},\sigma_{z}$ of the individual fermions as
$$\begin{array}{c} S_{x,y,z}=\frac{1}{2}\sigma_{x,y,z}⊗I+I⊗\frac{1}{2}\sigma_{x,y,z}\;\mathrm{and}\;S^{2}=S_{x}^{2}+S_{y}^{2}+S_{z}^{2}\;, \end{array}$$
written in the basis of products of single particle eigenstates along *z*, namely |++〉,|+−〉,|−+〉,|−−〉. However, some of these vectors are not eigenstates of $S^{2}$. A common eigenbasis to $S^{2}$ and $S_{z}$, written in terms of the product of single particle eigentstates, is $\left\{|++\rangle ,\frac{1}{\sqrt{2}}\left(|+-\rangle +|-+\rangle \right),\frac{1}{\sqrt{2}}\left(|+-\rangle -|-+\rangle \right),|--\rangle \right\}$. In particular, we will now focus on the subspace with spin 0 along the *z*-axis (m=0), i.e., the subspace generated by the two Bell states
$$\begin{array}{c} |\Psi^{+}\rangle =|\xi_{2,1,0}\rangle =\frac{1}{\sqrt{2}}\left(|+-\rangle +|-+\rangle \right)\;\;|\Psi^{-}\rangle =|\xi_{2,0,0}\rangle =\frac{1}{\sqrt{2}}\left(|+-\rangle -|-+\rangle \right)\;, \end{array}$$
which are entangled states and eigenstates of $S_{z}$ with m=0. We can then readily compute the spin-entropy for these states from Theorem 1 to be S=ln2π≈1.837. These Bell states, known to be maximally entangled states, have the lowest possible spin-entropy.

### 3.6. Qubit Entropy

An arbitrary state of a qubit can be written as a linear combination of the Pauli matrices, which provide a basis for all 2×2 self-adjoint matrices. A key difference between spin 12 particles and qubits is that, while all fermion particle systems must satisfy the Pauli exclusion principle, qubits can be constructed without it. For example, if a qubit represents the two possible states of an electron in a hydrogen atom, being in a specific excited state or the ground state, then two nearby hydrogen atoms can both be in the same state (both in the excited state or both in the ground state) or in a superposition of these states. When the two atoms are close enough to each other, it is also possible for them to be in a superposition of the ground state and excited state and entangled with each other. More broadly, it is possible for qubits to occupy any possible state in Hilbert space and to be transformed from one state to any other. For example, when building qubits technology, the CNOT gate [24,25] can disentangle any Bell entangled two qubit-state and also convert a disentangled qubit-state into a Bell entangled qubit-state.

Representing a qubit state in the basis |1〉,|0〉 (analogously to the spin *z* component up and down) leads to representing it by |10〉,|01〉 the two qubits disentangled states. We wish to define a qubit-entropy and to distinguish it from spin-entropy, and inspired by the 0,1 basis, we adopt the logarithm base 2 for qubits, i.e., we define the qubit-entropy to be

**Definition** **2.**
*SNqubits=1log2eSNs=12, where N is the number of qubits.*


### 3.7. Two Qubits Entanglement

Writing the disentangled states, |10〉,|01〉, in terms of the eigenstates of $S^{2},S_{z}$$|10\rangle =\frac{1}{\sqrt{2}}|\Psi^{+}\rangle +\frac{1}{\sqrt{2}}|\Psi^{-}\rangle$ and $|01\rangle =\frac{1}{\sqrt{2}}|\Psi^{+}\rangle -\frac{1}{\sqrt{2}}|\Psi^{-}\rangle$, we can evaluate the qubit-entropy to be
S2qubits(|10〉)=S2qubitsz(|10〉)+S2qubitsz⊥(|10〉)=1−∫12π1+cosϕln12π1+cosϕ(dϕ≈1log2e2.224.The qubit-entropy for |01〉 is the same. Clearly, the entropy for the disentangled two qubits is larger then the entropy for the Bell entangled states, which is $S_{2\;\mathrm{qubits}}(\frac{1}{\sqrt{2}}\left(|10\rangle \pm |01\rangle \right)\approx \frac{1}{\log_{2}e}1.837$.

#### 3.7.1. Comparison with the von Neumann and Werhl Entropies

For pure entangled states, von Neumann entropy is zero. However, one can trace out some states to evaluate von Neumann entropy of the remaining mixed states. Work exists exploring the information content of pure entangled physical systems employing von Neumann entropy, e.g., [26,27,28,29]. The required choice of basis functions is the product of one-particle eigenstates |+〉|−〉,|−〉|+〉, so that one can trace out one-particle spin eigenstates. Thus, in this basis, the Bell entangled states are represented as $|\Psi^{\pm }\rangle =\frac{1}{\sqrt{2}}|+\rangle |-\rangle \pm \frac{1}{\sqrt{2}}|-\rangle |+\rangle$, and after tracing out any of the two particles states the probabilities of the resulting mixed state are $P_{+}=\frac{1}{2}$ and $P_{-}=\frac{1}{2}$ with von Neumann entropy
$$\begin{array}{c} S_{2\mathrm{qubits}}^{\mathrm{vN}}\left(|\Psi^{\pm }\rangle \right)=1\;, \end{array}$$
while the disentangled states |+〉|−〉,|−〉|+〉, after one state being traced out, yield a single pure state with zero von Neumann entropy. Thus, disentangled states have lower von Neumann entropy then entangled states.

Wehrl’s entropy for such a mixed state was evaluated by [30] to conclude that, like von Neumann entropy, disentangled states minimize Wehrl entropy.

In contrast, the spin-entropy (and qubit-entropy) produce opposite evaluations of the entangled and disentangled states, being maximum at disentangled states and minimum at Bell states (the eigenstates of $S_{z},S^{2}$ for m=0). The spin-entropy and qubit-entropy also have an extra term due to the randomness of the conjugate ϕ variables in the chosen *z*-direction.

#### 3.7.2. Werner State

Werner state [31] is a state defined via a density matrix $\rho_{AB}$ that satisfies $\rho_{AB}=\left(U⊗U\right)\rho_{AB}\left(U^{†}⊗U^{†}\right)$ for every unitary operator *U* acting separately on both *d*-dimensional Hilbert subspaces *A* and *B*. For two qubits, corresponding to d=2, the density matrix in the basis {|11〉,|10〉,|01〉,|00〉} can be written as
$$\begin{array}{cc} W_{AB}^{\left(p,2\right)} & =\frac{p}{3}|11\rangle \langle 11|+\frac{3-2p}{6}|10\rangle \langle 10|+\frac{-3+4p}{6}|10\rangle \langle 01|\\ \; & \;+\frac{-3+4p}{6}|01\rangle \langle 10|+\frac{3-2p}{6}|01\rangle \langle 01|+\frac{p}{3}|00\rangle \langle 00|\;, \end{array}$$
where 0≤p≤1. Werner states are entangled for $p<\frac{1}{2}$ and separable for $p\geq \frac{1}{2}$. Wooters and colleagues [32,33] provide a method to evaluate an entropy of such state. They consider all possible pure state decompositions of $W_{AB}^{\left(p,2\right)}$. Each decomposition gives an entropy being the average of each pure state entropy according to the decomposition probabilities, and where each pure state entropy is the von Neumann entropy after tracing out one state. Finally, the entropy $W_{AB}^{\left(p,2\right)}$ is then defined as the minimum of such entropy over all possible pure states decomposition. Wooters calculated this entropy to be an increasing monotonic function according to the entanglement; the larger the entanglement, the larger is the entropy.

Our proposed qubit-entropy for the general case of mixed states is (7) and does not require any optimization. In order to evaluate Werner state qubit-entropy, we must first write it in the basis of the eigenstates of two-qubits spin matrices, namely, $\left\{|11\rangle ,\frac{1}{\sqrt{2}}\left(|10\rangle +|01\rangle \right),\frac{1}{\sqrt{2}}\left(|10\rangle -|01\rangle \right),|00\rangle \right\}$. Note that this basis decomposition of a two-qubit state will not keep the Werner density invariant. Instead, the Werner density in matrix form is
$$\begin{array}{c} W_{AB}^{\left(p,2\right)}=\left(\begin{array}{cccc} \frac{p}{3} & 0 & 0 & 0\\ 0 & \frac{p}{3} & 0 & 0\\ 0 & 0 & 1-p & 0\\ 0 & 0 & 0 & \frac{p}{3} \end{array}\right)\;. \end{array}$$

The qubit-entropy for this decomposition, following (7), is then
$$\begin{array}{cc} S_{2\mathrm{qubit}}^{M}\left(W_{AB}^{\left(p,2\right)}\right) & =-\left[p\log_{2}p+\left(1-p\right)\log_{2}\left(1-p\right)\right]+p\log_{2}3+\log_{2}2\pi \;, \end{array}$$
where the first term is the entropy of a binary coin with bias *p*, the second term is a linear term in *p*, and the last one is the qubit-entropy of the weighted sums of the pure states. The qubit-entropy reaches its maximum at $p=\frac{3}{4}$ when $W_{AB}^{\left(p=\frac{3}{4},2\right)}=\frac{1}{4}I$ is a totally mixed state and reaches its minimum at p=0 when it is in the pure entangled state $W_{AB}^{\left(p=0,2\right)}=\frac{1}{2}\left(|10\rangle -|01\rangle \right)\left(\langle 10|-\langle 01|\right)$. This is in contrast with Wooter’s evaluation based on von Neumann entropy, where the larger the entanglement, the larger the entropy.

### 3.8. Triplets

Given a triplet set of spin 12 particles, the total-spin matrices $S_{x,y,z}$ in the basis of products of single particle eigenstates are
$$\begin{array}{cc} S_{x,y,z} & =\sigma_{x,y,z}⊗I⊗I+I⊗\sigma_{x,y,z}⊗I+I⊗I⊗\sigma_{x,y,z}\;, \end{array}$$
where the Pauli matrices are written in the single particle *z*-axis eigenvectors basis. A common basis for both $S_{z}$ and $S^{2}$ is given by the vectors [34]
(17)$$\begin{array}{cc} |\xi_{\frac{3}{2},\frac{3}{2}}\rangle & =|+++\rangle ,\\ |\xi_{\frac{3}{2},\frac{1}{2}}\rangle & =\frac{1}{\sqrt{3}}\left(|++-\rangle +|+-+\rangle +|-++\rangle \right),\;\\ |\xi_{\frac{3}{2},-\frac{1}{2}}\rangle & =\frac{1}{\sqrt{3}}\left(|--+\rangle +|-+-\rangle +|+--\rangle \right),\;\\ |\xi_{\frac{3}{2},-\frac{3}{2}}\rangle & =|---\rangle ,\\ |\xi_{\frac{1}{2},\frac{1}{2}}\left(\theta_{+}\right)\rangle & =\frac{\cos\theta_{+}}{\sqrt{2}}|+\rangle ⊗\left(|+-\rangle -|-+\rangle \right)+\frac{\sin\theta_{+}}{\sqrt{2}}\left(|+-\rangle -|-+\rangle \right)⊗|+\rangle ,\\ |\xi_{\frac{1}{2},-\frac{1}{2}}\left(\theta_{-}\right)\rangle & =\frac{\cos\theta_{-}}{\sqrt{2}}|-\rangle ⊗\left(|+-\rangle -|-+\rangle \right)+\frac{\sin\theta_{-}}{\sqrt{2}}\left(|+-\rangle -|-+\rangle \right)⊗|-\rangle \;, \end{array}$$
where the parameters $\theta_{+},\theta_{-}\in \left[0,2\pi \right)$ characterize the degeneracy of the subspace of spin magnitude s=1/2, a four-dimensional subspace with only two eigenvalues.

#### Maximally Entangled States

According to our spin-entropy, the minimum ln2π is associated with the eigenstates (17), i.e., entangled states $|\xi_{\frac{3}{2},\frac{1}{2}}\rangle ,|\xi_{\frac{3}{2},-\frac{1}{2}}\rangle$ and entangled subspaces $|\xi_{\frac{1}{2},\frac{1}{2}}\left(\theta_{+}\right)\rangle ,|\xi_{\frac{1}{2},-\frac{1}{2}}\left(\theta_{-}\right)\rangle$. We propose that the minimum entropy be the criterion to define *maximally entangled states* among all the entangled states. Thus, the entangled state W [35], namely $|\psi_{W}\rangle =\frac{1}{\sqrt{3}}\left(|+--\rangle +|-+-\rangle +|--+\rangle \right)$, is a *maximally entangled state* since it is the eigenstate $|\xi_{\frac{3}{2},-\frac{1}{2}}\rangle$, and thus has the lowest entropy.

In contrast, consider the entangled state GHZ [36], $|\psi_{GHZ}\rangle =\frac{1}{\sqrt{2}}\left(|+++\rangle +|---\rangle \right)$. Clearly, it is not an eigenstate of either $S^{2}$ or $S_{z}$. The entropy will be larger than for all entangled eigenstates of $S^{2},S_{z}$ and thus it is not a *maximally entangled state*.

Ranking entangled states according to the spin-entropy generates an information content evaluation that may be helpful when devising quantum physical processes. The lower the entropy, the further away is the state from decoherence.

## 4. Conclusions

The concept of spin-entropy in a spin phase space is proposed. The spin phase space of a particle is defined via the already existing Geometric Quantization method that quantizes a sphere surface. The conjugate operators associated with the spherical polar representation do not commute, yielding the uncertainty principle for the spin values in phase space. The states in spin phase space are the simultaneous projections of a spin state onto the *z*-axis eigenstates and onto the polarization angle states, generating the plane perpendicular to the *z*-axis. The spin-entropy captures the randomness present in the spin state for a specified *z*-axis. The *z*-axis for a system of fermions is defined as the direction of the magnetic field present in the system, and it can vary over time. In the case of a photon, the *z*-axis is the direction of the propagation, where the helicity is defined. The formulation is general for a system of many spin particles and extends to quantum fields. We studied spin-entropy for single particles with spin 12, spin 1, photons, and for two and three entangled fermions of spin 12.

We have examined entangled states not only for the spin particles but also for qubits, where all possible two qubits are allowed. We then defined qubit-entropy as an extension of spin-entropy to qubits to the $\log_{2}$ base. Bell’s entangled qubit states that are eigenstates of the $S_{z},S^{2}$ operators have lower qubit-entropy than that of the product of one qubit states (disentangled qubit states). In contrast, the von Neumann entropy and Wehrl entropy are maximized at Bell’s entangled qubit states and minimized at disentangled qubit states. This analysis was also extended to Werner states.

We then analyzed some quantum states of three fermion of spin 12, also applicable to qubits, and suggested that maximum entangled states should be defined by the minimum spin-entropy value or qubit-entropy value. The lower the entropy of an entangled state, the larger the entanglement. 

## Figures and Tables

**Figure 1 entropy-24-01292-f001:**
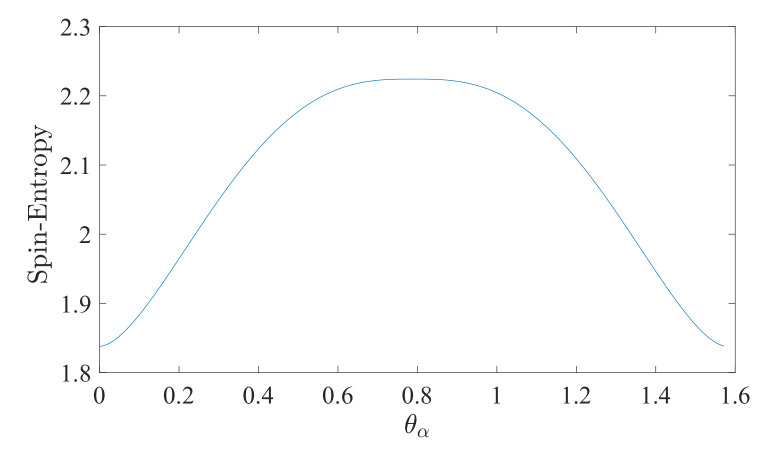
A plot of spin-entropy (11) vs. $\theta_{\alpha }\in \left[0,\frac{\pi }{2}\right]$, for s=12.

**Figure 2 entropy-24-01292-f002:**
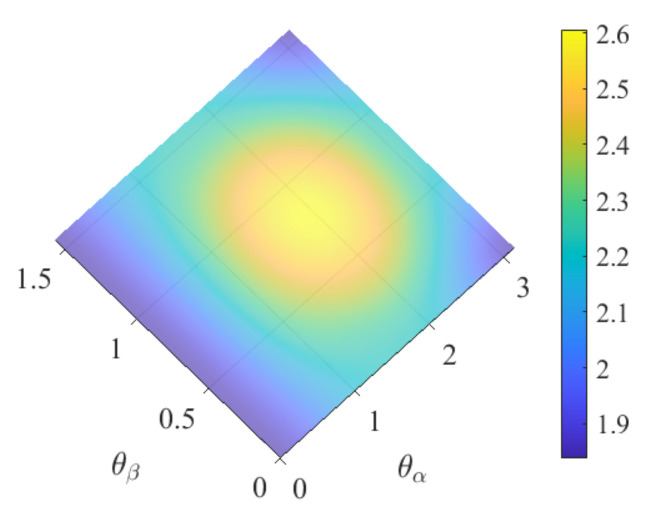
Spin-entropy for s=1 (15) vs. $\left(\theta_{\alpha },\theta_{\beta }\right)$. Note that, for $\theta_{\alpha }=0$ and for $\theta_{\alpha }=\frac{\pi }{2}$, $\theta_{\beta }=0,\frac{\pi }{2}$, describing the three eigenstates along the *z*-axis, the spin-entropy reaches its minimum value 1.838, approximating ln2π.

## Data Availability

The plots of figures were generated using Matlab program that the authors will make available on the first author’s website.

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
