# Peer review of "Spin Entropy"

_entropy, 2022, doi:10.3390/e24091292_

Round 1
Reviewer 1 Report
Comment in the attachment

Reviewer 2 Report
This paper proposes a new expression for the entropy of spin systems, focussing on the randomness of outcomes of spin measurements rather than on the difference between pure and mixed states (in the way of the von Neumann entropy). It is a technically competent piece of work containing interesting results.
It seems to me that the paper would be stronger, though, if more attention were paid to the question of why this approach is to be preferred over other approaches, in particular why this approach is better than the one using the vN entropy. Why is it advisable to use this new entropy for defining maximal entanglement? Entanglement is standardly defined as a property of states in Hilbert space; but the authors seem to change this definition into one involving spread in measurement results. This change is in need of justification, in my opinion. Is the claim perhaps that the statistics of measurement results gives empirical access to entanglement in a way that abstract Hilbert space considerations cannot provide?
In addition: In section 3.5.1 a symmetric two-fermion spin state is considered (so that the spatial part of the state must be anti-symmetric). Via an interaction with the environment this state is "disentangled", so that a product spin state results. Isn't this in violation of the Pauli exclusion principle? Permutation results in an orthogonal state. This process seems impossible, but perhaps I am overlooking something?
Finally, some more minor remarks. In the introduction it is said that for single particles the vN entropy is always zero. This is incorrect of course, single particles can be in mixed states. It is also said that entangled states tend to disentangle. This is not completely right, it seems to me: interactions generally lead to more entanglement, but it is true that entanglement spreads out over the environment. It is also misleading to say that for any entangled state the vN entropy is zero. Mixed states can be entangled. In section 2 it is said that joint knowledge of x and y spin is possible. This suggests that the x and y values are jointly well-defined, but that there are practical limitations to our knowledge of them. I think that the modern consensus is that this is an incorrect characterization of the situation.
Round 2
Reviewer 2 Report
The paper has improved and I have no objection against publication.